



# Towards vertical wind and turbulent flux estimation with multicopter UAS

Norman Wildmann[1] and Tamino Wetz[1]

[1]Deutsches Zentrum für Luft- und Raumfahrt e.V., Institut für Physik der Atmosphäre, Oberpfaffenhofen, Germany

**Correspondence:** Norman Wildmann (norman.wildmann@dlr.de)

**Abstract.** Vertical wind velocity and its fluctuations are essential parameters in the Atmospheric Boundary Layer (ABL) to determine turbulent fluxes and scaling parameters for ABL processes. The typical instrument to measure fluxes of momentum and heat in the surface layer are sonic anemometers. Without the infrastructure of meteorological masts and above their typical heights, in-situ point measurements of the three-dimensional wind vector are hardly available. We present a method to obtain the three-dimensional wind vector from avionic data of small multicopter unmanned aerial systems (UAS). To achieve a good accuracy in both, average and fluctuating parts of the wind components, calibrated motor thrust and measured accelerations by the UAS are used. In a validation campaign, in comparison to sonic anemometers on a 99-m mast, accuracies below $0.2 \, \mathrm{m\,s^{-1}}$ are achieved for the mean wind components and below $0.2 \, \mathrm{m^2 s^{-2}}$ for their variances. The spectra of variances and covariances show good agreement with the sonic anemometer up to 1 Hz temporal resolution. A case study of continuous measurements in a morning transition of a convective boundary layer with five UAS illustrates the potential of such measurements for ABL research.

## 1 Introduction

The nature of turbulence in the atmospheric boundary layer (ABL) is multifaceted and depends on internal and external parameters such as thermal stratification and shear, surface heat fluxes and surface roughness. Turbulent fluxes of momentum and heat are the main driver for mixing in the ABL and thus determine its diurnal cycle. Determining turbulent fluxes requires measuring the three-dimensional wind vector and temperature with a temporal resolution that covers all turbulent scales that contribute significantly to the process. The eddy covariance (EC) method is a way to determine the momentum fluxes from the covariances of the wind components $\overline{u'w'}$, $\overline{v'w'}$, $\overline{u'v'}$ and the heat flux from the covariance between potential temperature and vertical wind $\overline{\theta'w'}$ (Aubinet et al., 2012). From the fluxes, scaling parameters for the ABL such as shear stress $u^*$, convective velocity scale $w^*$ as well as Obukhov length $L$ and Richardson number $Ri$ can be derived (Stull, 1988) which are essential for a profound understanding of boundary-layer processes. It is thus evident that the measurement of vertical wind velocity and its fluctuations is crucial.

In practice, the most prominent instrument to measure turbulent fluxes in the ABL is the three-dimensional sonic anemometer (Kaimal and Businger, 1963) which has undergone extensive development and commercialization since the 1990s and is nowadays the essential part of every surface energy balance station (Mauder and Zeeman, 2018). Operating sonic anemometers is



mostly limited to the surface layer Baldocchi et al. (2001) or requires meteorological masts to reach heights up to 300 m above ground which involves significant costs and infrastructure (Wolfe and Lataitis, 2018). To obtain representative quantities for a certain area with single point measurements, assumptions of homogeneity and ergodicity need to be made. It has recently been shown that coherent structures such as secondary circulations can violate the ergodicity assumption and lead to errors

in the EC method (Mauder et al., 2020; Morrison et al., 2022). Only few experiments exist where multiple sonics have been installed over a wide area and heights above the surface layer, due to the large logistical effort. Some examples are the Great Belt Coherence experiment over water (Mann et al., 1991) and the Perdigão 2017 experiment in complex terrain (Fernando et al., 2019).

One possibility to obtain area-averaged fluxes are airborne measurements with research aircraft incorporating flow probes and

fast temperature sensors. Such measurements can cover many square kilometers in a relatively short amount of time, but also require assumptions of stationarity throughout the flight which can lead to sampling errors (Mahrt, 1998). To overcome limitations of manned aircraft to fly at low altitudes and at lower costs, fixed-wing unmanned aircraft systems (UAS) have been deployed in the past to measure turbulent fluxes (van den Kroonenberg et al., 2008; Wildmann et al., 2015).

To combine the benefits of the flexible deployment of UAS and allow point measurements instead of area-averaged measure-

ments, multicopter UAS are a possible solution. Wetz et al. (2021) showed that such systems can be deployed in small fleets and obtain distributed horizontal wind measurements with a good accuracy. They become increasingly popular for vertical profiling of the ABL when temperature and humidity sensors are carried (Koch et al., 2018; Segales et al., 2020). The measurement of vertical velocity with such systems is however challenging due to the obvious distortion of the flow by the thrust of the rotors. Thielicke et al. (2021) showed that with a sonic anemometer installed on a multicopter in sufficient distance to

the rotors, measurements of vertical wind is feasible. However, such a design increases the complexity of the flight system, its weight and cost. In order to operate a fleet of small and lightweight multicopter UAS, we propose a method to obtain the three-dimensional wind vector with avionic data alone, including calibrated motor thrust measurement. This is the first method that allows vertical wind and turbulent flux measurement with multicopter UAS without an external sensor.

In Section 2, we describe the UAS and the methodology to filter sensor data, calculate the forces that act on the UAS and

calibrate the vertical velocity measurements. Section 3 describes the setting of the calibration and validation campaign at the boundary layer field site (Grenzschichtmessfeld, GM) Falkenberg of the German Meteorological Service (DWD). Section 4 contains the results of three-dimensional wind measurements, second-order statistics and the derivation of momentum fluxes in comparison to measurements of sonic anemometers on a 99 m mast. A case study with measurements during a morning transition of a convective boundary layer illustrates the potential of the method. We conclude and give an outlook on future

research in Sect. 5.



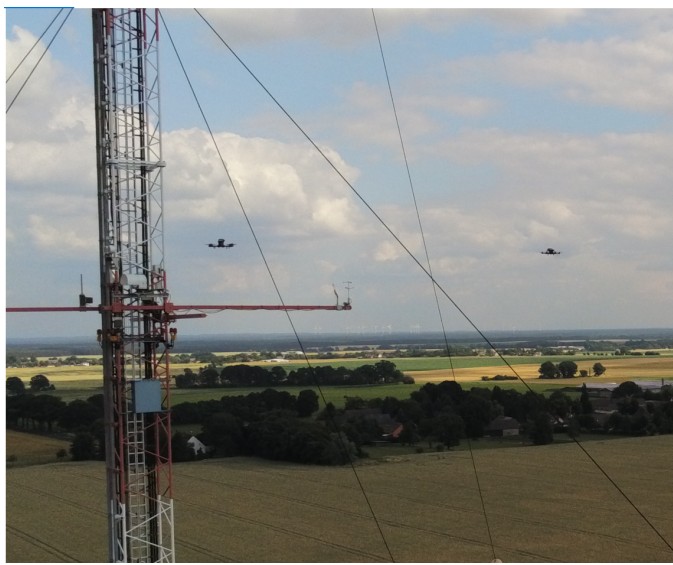

**Figure 1.** Two Holybro QAV250 UAS in flight. In the background the 99-m mast with a sonic anemometer on the boom at 50 m can be seen.

## 2 Methods

### 2.1 System description

We operate a fleet of 35 UAS of type Holybro QAV250 within the SWUF-3D project (Simultaneous Wind measurements with Unmanned Flight Systems in 3D). The main characteristics are described in Wetz et al. (2021). With a weight below 1 kg and a frame size of 0.25 m, the single UAS are small enough to be operated in the open category of the European Regulation 2019/947. Operation in a fleet with fewer pilots than UAS requires special permission from the flight authorities. A Pixhawk®4 Mini autopilot is used as the flight controller and allows access to all avionic data which is stored at high sampling rates on an internal SD card. The sampling rate depends on the type of data and is presented in more detail in Sect. 2.5. In the current configuration, flight times up to 17 minutes can be achieved. Figure 1 shows two UAS in flight.

### 2.2 Motor thrust calibration

In order to determine the forces that act on the UAS, it is important to know the actuator forces of the four rotors. There are no direct sensors for rotational speed, torque or even thrust in the system. However, the motor controller PPM (pulse-position modulation) signal as well as the battery voltage and current are measured by the autopilot. The PPM signal is directly related to the voltage which is applied to the motor and is in theory directly proportional to the revolution speed. The voltage at each rotor is

$$U_i = U_b \frac{s_i - s_{\min}}{s_{\max} - s_{\min}} \quad , \tag{1}$$



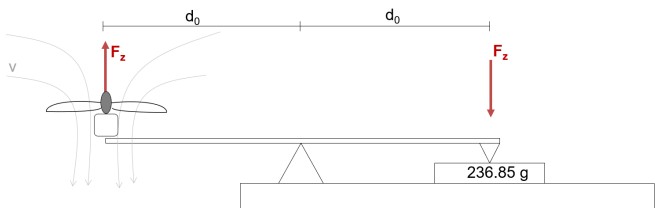

**Figure 2.** Schematic of the motor test stand. The rotor is producing thrust in the free stream and the counter-acting force of the equal length lever-arm is measured with a scale on a table.

with $U_b$ the battery voltage and $s_i$ the servo PPM command ($s_{\min}$ for zero and $s_{\max}$ for full throttle). It is important to include the battery voltage here, as it decreases throughout the flight. Assuming a linear relationship for the electric servo motors, the revolution speed of each motor can be determined as

$$\Omega_i = k_V U_i \quad .$$ (2)

The motor thrust $T_i$ can be calculated from the rotational speed $\Omega_i$, the rotor radius $R$, the rotor swept area $A = \pi R^2$, air density $\rho$ and the thrust coefficient $C_t$ (Leishman, 2016, Eq 2.31):

$$T_i = C_t \rho A \Omega_i^2 R^2$$ (3)

In order to determine this relationship experimentally, as well as the thrust coefficient of the rotor, which relates thrust to

revolution speed, a hardware-in-the-loop test stand for a single motor/rotor setup is used. A schematic of the setup is shown in Fig. 2. We can thus determine the motor constant $k_V$ and the thrust coefficient $C_t$ experimentally.

## 2.3 Quadrotor dynamics and rotor aerodynamics

To describe the motion of a quadrotor in three dimensions in its body-fixed frame of reference ($x$ to the front, $y$ to the right, $z$ downwards), the rigid body model as described in Wetz et al. (2021) can be used:

$$m(\ddot{x} + q\dot{z} - r\dot{y}) = -mg[\sin(\theta)] + F_x$$ (4)

$$m(\ddot{y} + p\dot{z} - r\dot{x}) = -mg[\cos(\theta)\sin(\varphi)] + F_y$$ (5)

$$m(\ddot{z} + p\dot{y} - q\dot{x}) = mg[\cos(\theta)\cos(\varphi)] + F_z - T \quad ,$$ (6)

with $m$ the mass of the UAS, $p$, $q$ and $r$ the gyroscopic terms, $g$ the acceleration due to gravity and $\theta$ and $\varphi$ the pitch and roll angle. The external forces $F$ in all three components are caused by wind. Eq. 6 contains the total actuator thrust $T$ which

counteracts all the external forces due to wind and gravity. The concept of operation for the UAS fleet is to hover at fixed positions, so that motions due to maneuvering can be neglected. Gyroscopic terms become very small and can be neglected in the following. In Wetz et al. (2021), only the forces in $x$-direction in Eq. 4 were considered, since the UAS yaws into the main wind direction and the wind speed magnitude can thus be approximated by this one dimensional simplification. In Wetz





and Wildmann (2022), the two-dimensional wind vector was determined, including Eq. 5 in the algorithm and improving the dynamic response by the inclusion of the acceleration terms. In order to measure the full three-dimensional wind vector, Eq. 6 needs to be included. The single motor thrusts $T_i$ can be obtained from calibration as described in Sect. 2.2 and can be summed to obtain the total acting force $T = \sum T_i$. With increasing wind speed, the quadrotor lift will not only be generated by the rotor thrust, but aerodynamic effects start to occur. In theory, these effects are well described if forward flight speed is much larger than the velocity which is induced by the rotor $v_i$ ($V_\infty >> v_i$). Glauert's high-speed approximation (Leishman, 2016) gives an equation for the total resulting thrust for a single-rotor helicopter:

$$T = 2\rho A v_i V_\infty \tag{7}$$

For the quadrotors used in this study, the induced velocity is on the order of $10\,\mathrm{m\,s^{-1}}$. That means that Glauert's approximation can not be used for hover flight and wind speeds of the same order of magnitude. Nevertheless, lift effects are observed by reduced power consumption and less required thrust in dependence of wind speed. We do thus include a lift term $F_L$ in Eq. 6 to account for this effect, but need to determine the relationship experimentally.

With the assumption of hover state and the additional term for aerodynamic lift, we can then solve Eq. 4-6 for the external wind forces, so that

$$
\begin{aligned}
F_x &= mg\sin(\theta) + m\ddot{x} & (8) \\
F_y &= mg\cos(\theta)\sin(\varphi) + m\ddot{y} & (9) \\
F_z &= -mg\cos(\theta)\cos(\varphi) + m\ddot{z} + T + F_L(F_x) \quad . & (10)
\end{aligned}
$$

## 2.4 Determination of the three-dimensional wind vector

To translate the forces that act on the quadrotor to wind speeds, a straight forward approach which was applied in Wetz et al. (2021) is the Rayleigh drag equation. It is however evident, that the complex aerodynamic system of a multicopter is not perfectly represented by the physical relationships for solid bodys. Wetz and Wildmann (2022) have therefore allowed a calibration curve which is a power law with a free exponent. With this approach, higher accuracies are achievable. We apply this method here for all three dimensions, so that the wind components in the body frame of the quadrotor are

$$
\begin{pmatrix} u_b \\ v_b \\ w_b \end{pmatrix} = \begin{pmatrix} c_x F_x^{b_x} \\ c_y F_y^{b_y} \\ c_z F_z^{b_z} \end{pmatrix} \quad , \tag{11}
$$

where the parameters $b$ and $c$ need to be calibrated for the respective direction. For the vertical direction, it is conceivable that upward and downward motion of the quadrotor relative to the air is subject to significantly different drag. For this reason, a distinction of cases is done for the z-component so that

$$
w_b = \begin{cases} c_{z\uparrow} F_z^{b_{z\uparrow}} & F_z \geq 0 \\ c_{z\downarrow} F_z^{b_{z\downarrow}} & F_z < 0 \end{cases} \quad . \tag{12}
$$





**Table 1.** Sensor data overview and intial raw sampling rates.

| type of data | variables | sampling rate |
|---|---|---|
| accelerometer data | $a_x, a_y, a_z$ | 250 Hz |
| vehicle attitude | $\varphi, \theta, \psi$ | 20 Hz |
| vehicle local velocities | $v_x, v_y, v_z$ | 10 Hz |
| actuator outputs | $s_i$ | 10 Hz |
| battery data | $U_b$ | 2 Hz |
| temperature and humidity | $T, q$ | 1 Hz |
| barometric pressure | $P$ | 75 Hz |

The final step to obtain the meteorological wind vector $\mathbf{U} = (\mathbf{u}, \mathbf{v}, \mathbf{w})$ is a rotation of the wind speeds from the body frame into the geodetic coordinates with the Euler angles of the quadrotor and a subtraction of the linear velocities $(v_x, v_y, v_z)$ which are measured by the autopilot on the basis of GNSS (global navigation satellite system) information:

$$
\begin{pmatrix} u \\ v \\ w \end{pmatrix} = \mathbf{R}(\varphi, \theta, \psi) \begin{pmatrix} u_b \\ v_b \\ w_b \end{pmatrix} - \begin{pmatrix} v_x \\ v_y \\ v_z \end{pmatrix} \quad , \tag{13}
$$

### 2.5 Data preparation and filtering

The raw sensor data of the UAS are stored on the SD card with different sampling rates, depending on the data type, but with a synchronized timestamp. Table 1 presents the raw data types and variables that are used for the wind calculation and their initial sampling rate. Reasons for the different sampling rates are the capabilities of the sensors and the requirements for the flight controller. Except for the battery data, all data are sampled with a rate of 10 Hz or higher. The raw data can be subject to significant noise in the high frequency range due to sensor noise or vibrations. Preconditioning steps are taken before further processing of the data to obtain clean and synchronized data. Accelerometer data is filtered with a Butterworth filter to prevent anti-aliasing effects from frequencies above 10 Hz. All data is then interpolated and synchronized to a common time step with $\Delta t = 0.1 \ s$. After this step, noise can still be recognized in accelerometer and actuator data, so that for the wind calculation, frequencies above 2 Hz are cut off with an FFT filter in the frequency domain.





## 2.6 Calculation of turbulence parameters

From the three-dimensional wind data, we can calculate momentum fluxes, friction velocity and turbulence kinetic energy (TKE). Momentum fluxes are calculated as the covariances between the wind components, i.e. $\overline{u'w'}, \overline{v'w'}$ and $\overline{u'v'}$. The friction velocity $u^*$ is calculated from the momentum fluxes according to Stull (1988):

$$u^* = \left( \overline{u'w'}^2 + \overline{v'w'}^2 \right)^{1/4} \tag{14}$$

TKE is defined as half of the sum of the variances of the three wind components:

$$TKE = \frac{1}{2} \left( \overline{u'}^2 + \overline{v'}^2 + \overline{w'}^2 \right) \quad . \tag{15}$$

## 3 Field experiment

A calibration and validation campaign for the SWUF-3D fleet was carried out in the framework of the FESSTVaL (Field Experiment on submesoscale spatio-temporal variability in Lindenberg) campaign in June/July 2021 (Hohenegger et al., in preparation) at the GM Falkenberg of the DWD. Figure 3 gives a map of the location with an indication of the measurement positions, including the location of the 99 m meteorological mast at the site. Two sonic anemometers (Metek UAS-1) are installed at 50 m and 90 m height which serve as the reference for calibration and validation. Their measurements are distorted by the tower for wind directions between 345° and 50° via north and are thus disregarded for such conditions.

Calibration flights were done with five UAS at 50 m and 90 m respectively, in close proximity ($\Delta x = 20$ m) to the mast (see black stars in Fig. 3 and Wetz and Wildmann (2022)). During a morning transition of a convective boundary layer on 28 June 2021, continuous measurements were done with two sets of five UAS stacked in five heights (10 m, 50 m, 90 m, 150 m, 200 m) at the location of the blue star in Fig. 3. The two sets were measuring alternately in close separation of 5 m and with a short overlap of one minute to guarantee continuous observations. Spatially distributed measurements (red stars) were performed to calculate coherence of the flow, but are not analysed in this study.

## 4 Results

### 4.1 Motor test stand calibration

With Eq. 1, the voltage at the motor can be estimated from the actuator outputs. Results from the hardware-in-the-loop experiment (Fig. 4a) reveal a nearly linear relationship between voltage and revolution speed. The coefficient that relates voltage to revolution speed is $k_V \approx 30.7 \, \mathrm{s^{-1} V^{-1}}$.

From Fig. 4b) we find that the thrust curve $T_i$ as a function of revolution speed $\Omega_i$ is best approximated with a polynom of second order. Such a behaviour is quite common and has been described before (Bangura et al., 2016)). We determine the





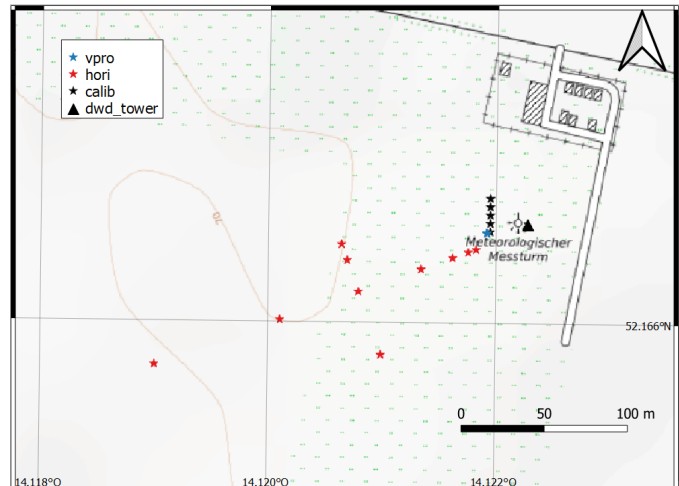

**Figure 3.** Map of the GM Falkenberg with the location of measurement points for the different flight patterns (stars) and the meteorological mast (black triangle). Background map ©OpenTopoMap contributors 2022. Distributed under a Creative Commons BY-SA License.

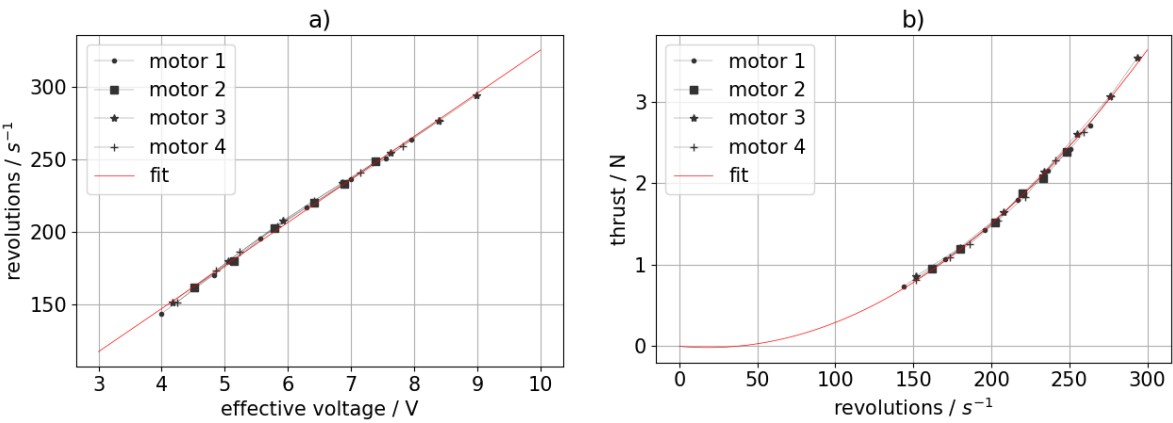

**Figure 4.** Calibration results of the motor test stand. a) The calibration curve of revolution speed versus applied voltage and b) the thrust coefficient curve. The fitted calibration curves are shown as red lines.

calibration curve as:

$$T_i = 4.6 \cdot 10^{-5} \Omega_i^2 - 1.68 \cdot 10^{-3} \Omega_i \quad . \tag{16}$$

### 4.2 In-flight wind vector calibration

For calibration of the dynamic equations as described in Sect. 2.3, we use UAS#12, because it was operating in most of the calibration flights (i.e. 14 flights, with a wind speed range of 1–8 m/s). The results of the calibration of the horizontal wind




components are presented in Wetz and Wildmann (2022). In this study, we focus on the calibration of the vertical velocity

component. This calibration is done in two steps. First, the lift force which reduces the necessary thrust in hover flight is determined as a function of the wind speed in $x$-direction. Second, thrust minus lift is calibrated against the vertical wind velocity as measured by the sonic anemometers at the 99-m mast.

### 4.2.1 Lift force

To estimate the lift force $F_L$, we analyse the measured thrust force during all calibration flights in dependency of the horizontal

wind. Horizontal wind is not known a priori, so we use the drag force in $x$-direction in the body frame $F_x$ as a proxy which depends on the wind speed as described by Eqs. 8 and 11. It is not possible in a field experiment to perfectly decouple the lift effects from the wind in $x$-direction from the drag and lift in $z$-direction. For calibration purposes, we assume that the $x$ and $z$-component of the wind are uncorrelated and that vertical wind is normally distributed around zero. To reduce the scatter, a 10 s moving average is applied to the data. Figure 5 shows the point cloud of lift versus drag force in x-direction for all calibration

flights of UAS#12. The red line gives a polynomial fit. It can be seen that significant lift is generated up to $F_x \approx 1$ N lift, before it abruptly decreases, as it is typical for stall conditions. This force in x-direction corresponds to a horizontal wind speed of approximately $8 \, \mathrm{m \, s^{-1}}$. Beyond that wind speed, only few data are available which yields a high uncertainty of the calibration at higher wind speed conditions. The determined polynomial for the lift correction is:

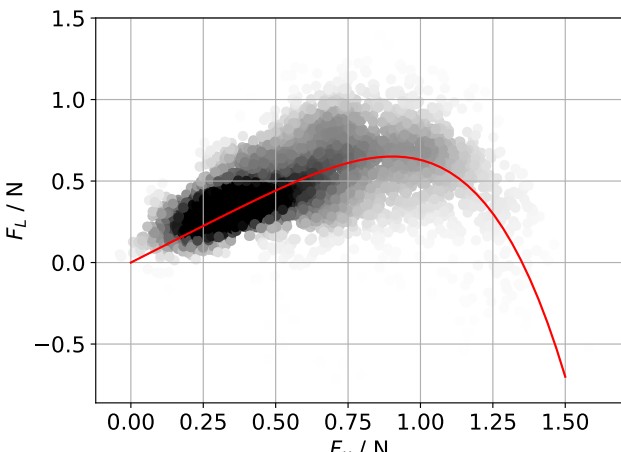

**Figure 5.** Calibration curve of thrust generated by lift $F_L$ versus drag force in $x$-direction $F_x$ in the body frame. Greyscale represents the probability of occurence.

$$F_L = T_0 + 0.9 F_x - 0.27 F_x^5 \quad , \tag{17}$$

where $T_0$ is a thrust offset value that can depend on changes in UAS mass, changes in motor thrust coefficients or unmodelled aerodynamic effects.





### 4.2.2 Thrust offset

We observe that the absolute value of thrust $T_0$ that is needed to maintain altitude in the hovering state varies between single flights and especially between individual UAS. The reasons can be differences in UAS mass, motor performance or aerody-
namic effects. At the present state of development and hardware robustness, we find that $T_0$ needs to be determined in each single flight as the average $\overline{T}$ over the whole flight for best performance. This is justified in flat terrain where zero average vertical velocity can be assumed in most cases. For cases with non-zero vertical velocity, determining $T_0$ from the average over a single flight can yield errors in the absolute value of $w$, but will have little effect on the variance, which is most important for the calculation of TKE and fluxes.

### 4.2.3 Vertical velocity

In the second step of the calibration, vertical wind is calibrated against thrust (including aerodynamic lift, see above) with data from the FESSTVaL field campaign. Figure 6 shows the scatterplot of all calibration flights of UAS#12. Since measurements of UAS and sonic anemometer are comparatively far apart ($\approx 20\,m$) and absolute values of vertical wind are rather small, it is not surprising that significant scatter is found in the comparison. It is nevertheless possible to fit an optimal curve according to
the theory of Eq. 18. The fitted equation is:

$$w_b \quad = \quad \begin{cases} 3.3|F_z|^{0.85} & F_z \leq 0 \\ -1.6|F_z|^{0.6} & F_z > 0 \end{cases} . \tag{18}$$

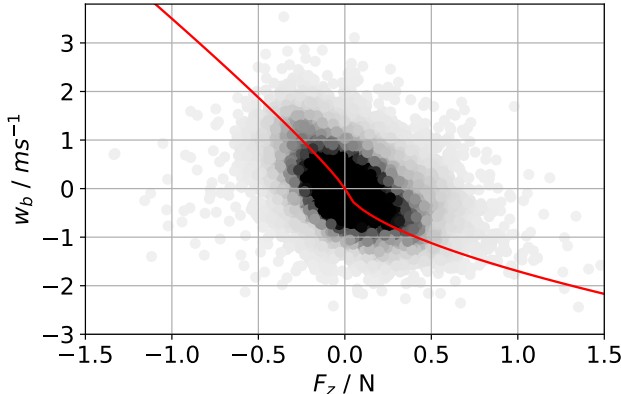

**Figure 6.** Scatter plot of vertical wind of the sonic anemometer versus the acting force on the UAS in the body frame. Greyscale represents the probability of occurence in the dataset. The red line shows the estimated calibration curve.





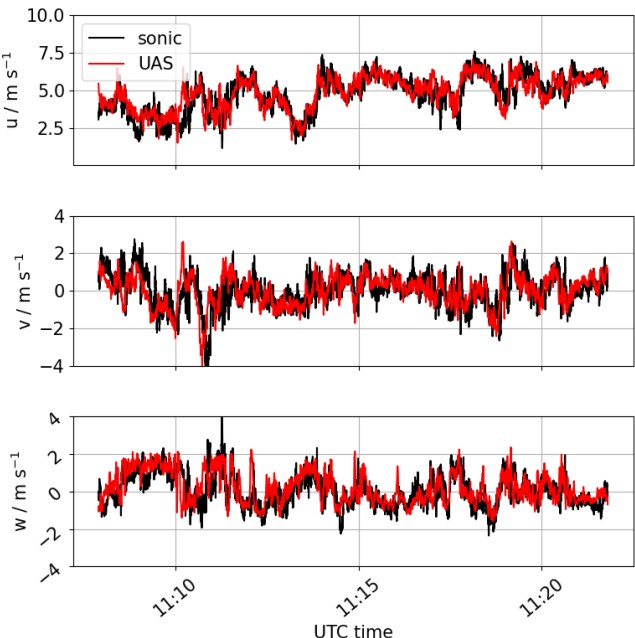

**Figure 7.** Example of time series of $u$, $v$ and $w$ for a flight on 02 July 2021 by UAS#13. UAS data (red) is compared to the sonic anemometer at the same height (black).

## 4.3 Wind vector validation

The parameters that are determined for UAS#12 are now applied to the whole fleet. We now want to compare the wind

components $u$, $v$ and $w$ in the geodetic coordinate system according to Eq. 13. Nevertheless, it is more meaningful to rotate the vector horizontally into the main wind direction for both, sonic anemometer and UAS in order to evaluate the different calibration coefficients. The time series in Fig. 7 demonstrates the quality of the wind retrieval in all three dimensions and at a high resolution for a flight in a turbulent ABL with horizontal wind speeds ranging from $u = 2.5 - 7.5 \, \mathrm{m \, s^{-1}}$. Figure 8 gives scatter plots of the comparison of flight-averaged wind components, their variances and the three covariances $\overline{u'w'}$, $\overline{v'w'}$

and $\overline{u'v'}$. Since the wind vector is rotated into the main wind direction during the flight time, the lateral wind component $v$ is zero by definition (Fig. 8b). Flight-averaged vertical velocities are smaller than the uncertainty of the measurement so that a linear regression does not yield a good correlation ($P$-value for test of null-hypothesis $P > 0.1$). The root-mean squared (RMS) deviation is $\epsilon_w = 0.25 \, \mathrm{m \, s^{-1}}$. Better parameters to evaluate the quality of the wind retrieval algorithm are the variances of the wind components in lateral and vertical direction. For those, the linear regressions show a very good correlation between UAS

and sonic anemometer ($R^2 > 0.9$, Fig. 8e,f). The covariance measurements are a good indicator if an independent measurement of horizontal and vertical wind can be guaranteed with the proposed method. The high correlation proves the validity of the method and the RMS deviation of approximately $\varepsilon = 0.1 \, \mathrm{m^2 s^{-2}}$ for all directions can be considered the prevailing uncertainty





of the UAS measurements (Fig. 10g,h). The lower $R^2$-value for $\overline{v'w'}$ is likely due to the imperfect rotation into wind direction for both, UAS and sonic anemometer which introduces larger relative errors for the overall small values.



**Figure 8.** Scatter plots of flight-averaged UAS measurements versus sonic anemometer for $u$, $v$ and $w$ (a-c), as well as the corresponding variances (d-f) and covariances (g-i).





## 4.4 Validation of resolved turbulence scales

A closer evaluation of the resolved turbulence scales by the UAS can be achieved with the analyses of power spectra of the three wind components in comparison to the sonic anemometer. We choose two representative flights of UAS#12, one in low wind and low turbulence conditions in the morning of the 26 June (blue), the other in a turbulent convective boundary layer on 2 July (red). The spectra of the three wind components in Fig. 9 show that an inertial subrange of locally isotropic turbulence with a slope of -5/3 (dashed grey line) can be observed in both cases. For the low turbulence case, the spectra as measured by the UAS deviates earlier from the curve as measured by the sonic anemometer than in the high turbulence case. It can be read from the plots that frequencies above 1 Hz and spectral power below $10^{-2}\,\mathrm{m^2\,s^{-2}\,Hz^{-1}}$ can barely be resolved by the UAS, but are dominated by noise. This effect is also observed for the cospectra of the wind components (Fig. 10). For conditions with low turbulence, the noise contributions are limiting the resolution of variance and covariance measurements with the UAS. From the noise level, we estimate this limit in resolution for the variances and covariances to $\approx 0.1\,\mathrm{m^2 s^{-2}}$ which is almost equal to the RMSE that was determined in Sect. 4.3.

## 4.5 Momentum flux during a morning transition

A good case to study the validity and capability of the turbulence and flux measurements from the UAS is a transition phase of a convective boundary layer (CBL), which features some significant variability and increase of turbulence at constant heights. In the morning of 28 June 2021, conditions were present which allowed to observe a morning transition of a CBL which was mainly driven by surface heating. In the cloudless night, a stable nighttime inversion had developed and still no clouds were present during the period of observation from 0400 UTC to 0800 UTC. A steady increase of shortwave downward radiation from $\approx 100\,\mathrm{W\,m^{-2}}$ to $\approx 700\,\mathrm{W\,m^{-2}}$ was observed.

Continuous measurements during this period with five stacked UAS at the heights 10 m, 50 m, 90 m, 150 m and 200 m were achieved by launching two separate sets of five UAS in turns, with a short overlap of 60 s. The UAS were located close to the meteorological mast as indicated by the blue star in Fig. 3. In total, 23 flights with five UAS each, were performed in this pattern. Additionally, a sixth UAS was operating in close proximity ($\approx 5$ m apart), profiling the same height range with continuous ascents and descents. The ascents of these flights allow to obtain profiles of wind speed, wind direction, temperature and humidity with a high vertical resolution, but are not used for the turbulence retrieval. Figure 11 shows the results of the measurements during the morning transition. On the left, the time series of the UAS at 50 m are shown in comparison to the sonic anemometer at the mast at the same level. The data are post-processed with a 10-minute moving average for better visualization. Between 0730 UTC and 0745 UTC, data are missing because one UAS did not take off as planned. The dots in the plots show the 30-minute average with error bars representing the uncertainties that were estimated in Sect. 4.3. We find a good agreement between UAS and sonic anemometer for all variables. Around 0530 UTC, the mixed CBL is growing through the measurement height, wind becomes very small and thus the estimation of wind direction deviates by almost twenty degrees between UAS and sonic anemometer. Also, temperature measurements differ most during this period. TKE only picks up significantly after 0630 UTC with a large peak just before 0800 UTC which can be seen in both UAS and sonic anemometer.




The friction velocity $u^*$ is quite variable, also increasing after 0630 UTC.

In the right panel of Fig. 11, Hovmoeller diagrams show the transition period in dependency of time and height. From the

profiles of potential temperature, an estimate of the boundary-layer height as the height of the mixed layer below the lowest inversion is derived and shown as a black line. A low-level jet (LLJ) was present at 100 m above ground level (a.g.l.) which disappears as the mixed layer grows and is the reason for the observed higher wind speeds in the early morning at 50 m and above. It can be clearly seen how low turbulence and high wind speeds above the inversion are separated from low wind speeds and higher turbulence in the mixed layer. Despite the large number of individual UAS that were used to take these

measurements (i.e. 12 UAS), no significant biases or discontinuities can be found in the data, which is a good validation of the accuracy of the calibration.

## 5    Conclusions

In this work, we present a method to obtain the three-dimensional wind vector with small multicopter UAS. To obtain the vertical wind velocity, motor thrust is included in the equations of motion and the resulting forces that act on the UAS are

calibrated against sonic anemometers to obtain the wind velocity with an RMS deviation below $0.2\,\mathrm{m\,s^{-1}}$ in all three dimensions. We showed that the variance of the wind components can be obtained particularly well with a remaining RMS deviation below $0.2\,\mathrm{m^2\,s^{-2}}$ in all dimensions. The range and limits of the resolvable scales of turbulence are presented with spectra and cospectra of the wind components. It is evident that there is a noise limit in the temporal resolution of approximately 1 Hz and a limit for low turbulence conditions with spectral power density below $10^{-2}\mathrm{m^2\,s^{-2}\,Hz^{-1}}$. The fact that we could exercise the

calibration with a single UAS and apply it to a fleet of more than 20 UAS without a big loss of accuracy shows the robustness of the method and the applicability in practice. From these findings, it can be concluded that UAS of the presented size and weight are capable of resolving eddies of only few meters. Bigger and heavier UAS will necessarily disturb the flow more, so that a small and lightweight design is essential for a high resolution of wind measurements.

In the presented dataset, the range of horizontal wind velocities is limited to $\approx 8\,\mathrm{m\,s^{-1}}$. We see that above such wind speeds,

aerodynamic effects change and more studies will be necessary in future to extend the calibration to higher wind speeds. We remove a thrust bias for every individual flight in this study, because the assumption of zero average vertical wind speed can be made in the flat terrain in which the experiment was carried out, and this approach will lead to best estimates of vertical velocity variance. For conditions with a significant average vertical velocity component, different strategies will need to be developed in future, which may incorporate more robust hardware configurations, an integrated weight estimator in the Kalman

Filter or individual calibration of motor thrusts for each UAS. For both constraints, the obtained dataset from the FESSTVaL campaign can not be used to improve the calibration. In general, calibration with field data is prone to large uncertainties due to the complex ABL flow. The calibration dataset is thus subject to significant scatter and the polynomials that are used for the relationship between forces and wind speed are not strictly based on physical principles, as simplifying assumptions are made. In future, we plan to conduct wind tunnel experiments to study the aerodynamic effects in a broader range of horizontal

and vertical wind velocities and in more detail. Hardware modifications that allow hover flights indoor are currently prepared



for this purpose. We also prepare temperature sensors with a faster response time to be able to measure sensible heat flux in a broad range of atmospheric conditions in future.

We showed the potential of multicopter UAS measurements for fundamental ABL research with a case study during a morning transtition of a CBL. In this case study, the calibration was applied to 12 individual UAS and nevertheless, a seamless time series over 4.5 hours could be realized in six measurement heights. The evolution of wind speed and temperature, as well as TKE and shear stress is well captured in comparison to the reference from sonic anemometers throughout the transition period. This example demonstrates the advantage of operating a fleet of multiple UAS to obtain spatio-temporal in-situ data at flexible measurement points throughout the ABL. The possibilities to deploy such a system for ABL studies in complex flow such as wind farms or mountainous terrain are numerous and an observational gap can potentially be closed with this approach.

*Data availability.* Data is available from the authors upon request and will shortly be made available through the FESSTVaL data archive.

*Author contributions.* Both authors jointly developed and carried out the field experiment. NW developed the ideas to retrieve vertical velocity and carried out the motor calibration. TW developed the calibration for horizontal wind speed.

*Competing interests.* The authors declare that there are no competing interests.

*Acknowledgements.* We acknowledge the support of DWD and the GM Falkenberg in the conduction of the experiment. We thank May Bohmann and Josef Zink for their assistance in the execution of the experiments. Many thanks to Frank Holzäpfel for an internal review of the manuscript.





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





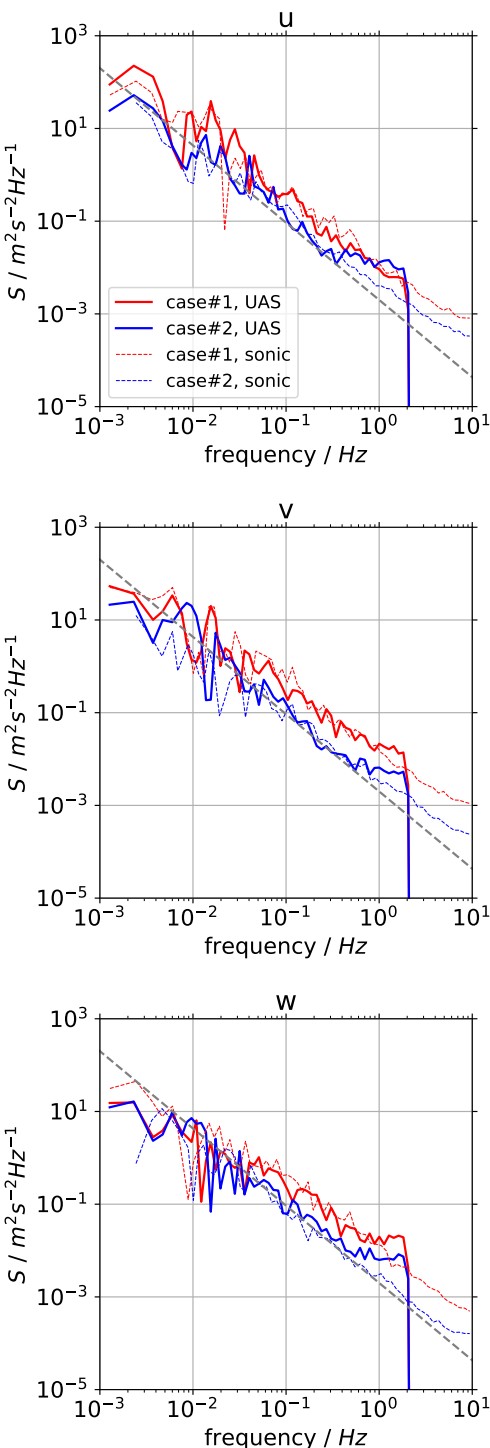

**Figure 9.** Comparison of power spectra of all three wind components between sonic anemometer and UAS.





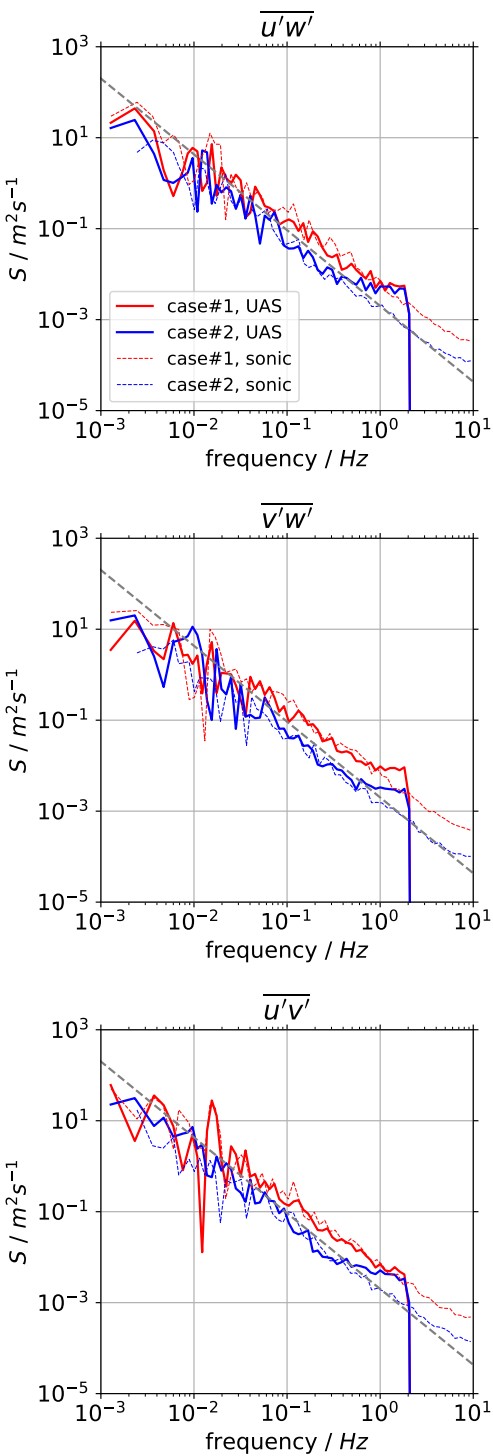

**Figure 10.** Comparison of power cospectra between sonic anemometer and UAS.







**Figure 11.** Time series of horizontal wind speed $U$, wind direction $\Psi$, temperature $T$, friction velocity $u^*$ and TKE at 50 m above ground level during the morning transition of 28 June 2021 in comparison between UAS and sonic (left). the continuous lines show 10-minute moving averages, whereas the dots give the 30-minute mean for each quantity. On the right, Hovmoeller diagrams for the observed height range from 10-200 m are shown. The black line indicates the height of the mixed layer below the lowest temperature inversion as observed from vertical profiles of potential temperature $\theta$. For wind speed $U$, wind direction $\Psi$ and $\theta$, continuous ascent vertical profiles from a single drone are shown in the background of the measurements from the hovering UAS at 10 m, 50 m, 90 m, 150 m and 200 m.