# Peer review of "Towards vertical wind and turbulent flux estimation with multicopter UAS"

_EGUsphere, 2022_

## Referee Comment (RC1)

The manuscript presented by Wildman and Weltz proposes a new method for estimating horizontal and vertical components of wind velocity, and respective time fluctuations. This area of research is critical for enabling new capabilities for increasing the spatiotemporal resolution of key observations in the lower atmosphere. However, additional validation is required to support the claims sustained by the authors. Therefore, I will withhold my recommendation for publication until the comments I have provided below are addressed by the authors.

The model presented in Eqs. (4) – (6) should be described as a point mass model instead of a rigid body model as it only accounts for translational forces and accelerations. A rigid body model account for sUAS rotational dynamics as well.

The authors assume gyroscopic terms to be small. To verify this assumption, the authors need to include a time history plot for each gyroscopic term.

In addition to reasons mentioned in line 89, absolute thrust can be influenced by the performance of GPS and barometer measurements, which are both used for hovering at a fixed height.

In Section 2.4, the authors mention employing the methods proposed by Weltz and Wildmann (2022) to calibrate curves for estimating wind velocity components in the body reference frame. However, curve fitting results are provided only for $w_b$ in Section 4.2.3. The $u_b$ and $v_b$ calibration results need to be presented as well for completeness.

With regards to assertion made in line 199 about optimality, an optimal fit is not always a good fit. The data scatter shown in Figure 6 is quite high, which minimizes the value of using a nonlinear model for relating forces to airspeed. The authors should compare the residual error from linear and nonlinear fits to justify the curve fitting formulation described in Eq (11).

In Section 4.3, the authors state that variance of the lateral and vertical wind components is a better parameter for evaluating the performance of the wind estimation retrieval. This approach is not sensible as two signals with different mean values can have a similar variance.

In Section 4.3, It is not clear why the authors argue that it is more meaningful to rotate UAS and sonic anemometer wind observation into a wind reference frame. As shown in Figure 8, doing so makes it difficult to validate estimates of $u$, $v$, and $w$, as well as respective time fluctuations, $u'$, $v'$, and $w'$. Additionally, validation of resolved turbulence scales need to be performed in a N-E reference frame.

The conclusions made in this manuscript are largely unsupported unless a meaningful validation of wind velocity components $u$, $v$, and $w$, as well as respective time fluctuations, $u'$, $v'$, and $w'$ are demonstrated with respect to an inertial reference frame.

---

## Referee Comment (RC2)

**Reviewer #2 Response**

The aim of this manuscript is to investigate the ability of small uncrewed aerial systems (UAS) to measure the 3 dimensional wind vector with enough accuracy to derive turbulent fluxes. The method by which the 3D wind vector is calculated is described in detail and is shown to perform well when compared to a co-located sonic anemometer. Additionally, a case study showing the applicability of the method is shown.

Overall, the manuscript is well written and devoid of any obvious errors in the presented methodology. Thus I have relatively few comments. With a few clarifications and additions, I think the manuscript would be suitable for publication.

**General Comments**

- Given that most of the analysis is centered around optimal fits of field data, which is inherently noisy, it would be informative to include some analysis of the uncertainty of these fits and the sensitivity of the final wind estimation to these uncertainties. This would also help to alleviate typical concerns about optimal fits not being based on physical principals.

**Specific Comments**

- The manuscript alludes to the fact that the wind-vane mode used in previous experiments was activated, but I don't see that explicitly stated anywhere.
- For the profiling UAS in the case study, are the same coefficients determined while hovering being used to estimate wind while ascending/descending? Bell et. al. (2020) showed that different coefficients are likely needed for an ascending UAS using a more rudimentary method for wind estimation. Would you expect the same here?
- Was there any correction necessary to match up the UAS and sonic time series? If so, how was this done?
- Were the winds from the sonic and UAS rotated into the mean wind independently of each other? In other words, was the sonic mean wind direction used to rotate to sonic winds, and vise versa? Or was one system used to determine the rotation for both?
- Though small, there does seem to be a consistent bias in wind direction in Fig 11. Would the bias be attributable to GPS errors? Was there a magnetometer calibration for the flight site?

**Technical Comments (representative, not comprehensive)**

- I recommend replacing 'unmanned aerial systems' with the more inclusive 'uncrewed aerial systems' or 'remotely piloted aircraft systems'

- The way the axis labels on the figures are formatted could cause some confusion at first. For example, on Fig 4 "revolutions / $s^{-1}$" could be interpreted as "revolutions per $s^{-1}$" instead of "revolutions with units of $s^{-1}$".
- Colorbars should be included on Figs 5 and 6.
- On Fig 7, it may be informative to also plot the differences of the time series.
- On Fig 11, the errorbars are pretty difficult to see since they line up with the grid lines. It may be worth putting the cap on the end of the error bars to make them more visible.

---

## Author Comment (AC1)

**Towards vertical wind and turbulent flux estimation with multicopter UAS**

Norman Wildmann[1] and Tamino Wetz[1]

[1]Deutsches Zentrum für Luft- und Raumfahrt e.V., Institut für Physik der Atmosphäre, Oberpfaffenhofen, Germany

**Correspondence:** Norman Wildmann (norman.wildmann@dlr.de)

**1 Review response**

We want to thank the reviewer for their careful and valuable review. We hope that we can clarify our analyses and clear out some of the concerns with our response.

**1.1 Review Comment 1**

1. *The model presented in Eqs. (4) – (6) should be described as a point mass model instead of a rigid body model as it only accounts for translational forces and accelerations. A rigid body model account for sUAS rotational dynamics as well.*
   We do not completely agree to this comment. A point mass only has three degrees of freedom, whereas a rigid body model has six degrees of freedom, including rotational terms. Strictly speaking, we are presenting the equations for the translational part of the rigid body model, which includes the rotational angles and gyroscopic terms, but we do not take the momentum equations into account. In a revised manuscript, we will be more clear about this.

2. *The authors assume gyroscopic terms to be small. To verify this assumption, the authors need to include a time history plot for each gyroscopic term.*
   The variance of the gyroscopic terms are two orders of magnitude smaller than the variances of the linear accelerations. This particularly applies as we are only considering the hovering parts of the flights. We give that information in Wetz and Wildmann (2022) and will also include it in this manuscript. We show a time history plot of the terms in Fig. 1. It shows the large difference between the terms, but we think that it is sufficiently well described in the text of the manuscript without the plot, which is not adding much information.

3. *In addition to reasons mentioned in line 89, absolute thrust can be influenced by the performance of GPS and barometer measurements, which are both used for hovering at a fixed height.*
   We agree that the performance of GPS and barometer measurements influence the thrust indirectly. Errors in the height measurement can lead to a control input of the autopilot to the motors, causing an ascend or descend of the quadrotor. This is definitely a potential source of error. Any measured vertical velocity of the UAS is subtracted from the wind measurement as described by Eq. 13. In general, the wind measurement depends largely on the accuracy of the Kalman

[Figure]

**Figure 1.** Example of time series of body-frame accelerations in $x$-,$y$- and $z$-direction for a flight on 02 July 2021 by UAS#13. Translational acceleration terms (blue) are compared to gyroscopic terms (red) for a hover period of the flight.

Filter solution for orientation angles and translational velocities. Our approach is to treat these uncertainties as unknown and validate the overall uncertainty of the system in the field. Determining the true accuracy of the state estimator in flight is extremely challenging, because no independent measurements are available. Sophisticated measurement setups with cameras or other external sensors could be set up in future, but are beyond the scope of this work.

4. *In Section 2.4, the authors mention employing the methods proposed by Weltz and Wildmann (2022) to calibrate curves for estimating wind velocity components in the body reference frame. However, curve fitting results are provided only for $w_b$ in Section 4.2.3. The $u_b$ and $v_b$ calibration results need to be presented as well for completeness.*

In this study, we focus on the vertical wind component. As written, we have shown in Wetz and Wildmann (2022) and Wetz et al. (2021) that horizontal components can be well measured and presented the calibration curves. We believe that another presentation of that calibration could be conceived as a repetition of old results. In Figure 8 of this manuscript we also give the validation of the horizontal wind component.

5. *With regards to assertion made in line 199 about optimality, an optimal fit is not always a good fit. The data scatter shown in Figure 6 is quite high, which minimizes the value of using a nonlinear model for relating forces to airspeed. The authors should compare the residual error from linear and nonlinear fits to justify the curve fitting formulation described in Eq (11).*

We agree that optimization is not always good, particularly with a noisy dataset. We agree that a linear fit could be
applied locally in the small range of observed vertical wind speeds. Actually, the exponential curve is very close to a
linear function in that range. Nevertheless, we think it is reasonable and justified to fit a curve with an increasing slope
at higher wind speeds, because this is what we would physically expect for drag. An exponential function is a natural
choice in our opinion, considering drag models such as the Rayleigh drag equation with an exponent of 0.5. As we state
in the conclusions, we believe that dedicated laboratory (wind tunnel) experiments should be performed in future to
improve the accuracy of the physical model. This is however not possible with the current dataset.

6. *In Section 4.3, the authors state that variance of the lateral and vertical wind components is a better parameter for
evaluating the performance of the wind estimation retrieval. This approach is not sensible as two signals with different
mean values can have a similar variance.*

From our point of view, the approach is sensible, because we do not look at two random variables without a causality.
We describe the physical process that causes the fluctuations. The control algorithm of the weather-vane mode of the
UAS will minimize the lateral wind component by yawing the UAS into the wind, which means that the lateral wind
component will always be very close to zero and thus the mean value can hardly be used to evaluate its accuracy. In
flat terrain, we will also mostly find average vertical wind speeds that are close to zero, so that an evaluation of the
performance of the algorithm is also quite unreasonable with mean values of $w$. Since we are interested in turbulence,
we think that looking at variances is not only sensible, but also important for the evaluation of the measurement system.

7. *In Section 4.3, It is not clear why the authors argue that it is more meaningful to rotate UAS and sonic anemometer wind
observation into a wind reference frame. As shown in Figure 8, doing so makes it difficult to validate estimates of $u$, $v$,
and $w$, as well as respective time fluctuations, $u'$, $v'$, and $w'$. Additionally, validation of resolved turbulence scales need
to be performed in a N-E reference frame.*

We will try to make the selection of the frame of reference more clear. For two reasons, looking at streamwise and lateral
wind direction is more meaningful than a N-E frame. First, from a measurement perspective of the UAS wind algorithm,
the streamwise velocity component reflects the rotation of the quadrotor in the pitch-axis and the lateral component in
the roll-axis. This is particularly interesting because the weather vane mode is a controller that minimizes the roll angle.
It is true that the mean value of $v$ becomes less meaningful, but the variance of $v'$ is important to check, as it reflects
the disturbance of the weather vane controller. Second, if we want to study turbulent structures, e.g. the coherence or
the length scales of large eddies, it is more meaningful to look at streamwise and lateral components than a N-E frame,
which is not related to the flow field. We thus do not agree that turbulence scales need to be validated in the N-E reference
frame.

8. *The conclusions made in this manuscript are largely unsupported unless a meaningful validation of wind velocity com-
ponents $u$, $v$, and $w$, as well as respective time fluctuations, $u'$, $v'$, and $w'$ are demonstrated with respect to an inertial
reference frame.*

As stated above, we disagree that an analysis in a N-E frame would be more meaningful. We believe that our conclusions

are based on a solid database, which is far beyond what has been presented from UAS measurements so far, given the large number of flights and atmospheric conditions in comparison to sonic anemometers on two height levels. We present calibration and validation for all the parameters of interest. As the reviewer suggests, we present here the results of the validation in the meteorological frame of reference and will include it in a revised manuscript:

[Figure]

**Figure 2.** Scatter plots of flight-averaged UAS measurements versus sonic anemometer for $u$, $v$ and $w$ (a-c), as well as the corresponding variances (d-f) and covariances (g-i) in the meteorological frame of reference.

The N-E components are a rotation of the results in the streamwise and lateral direction by the wind direction. It can be seen that RMSE and R-values are of the same magnitude as presented for the wind reference frame.

**References**

80    Wetz, T., Wildmann, N., and Beyrich, F.: Distributed wind measurements with multiple quadrotor unmanned aerial vehicles in the atmospheric boundary layer, Atmospheric Measurement Techniques, 14, 3795–3814, https://doi.org/10.5194/amt-14-3795-2021, 2021.

Wetz, T. and Wildmann, N.: Spatially distributed and simultaneous wind measurements with a fleet of small quadrotor UAS, Journal of Physics: Conference Series, 2265, 022 086, https://doi.org/10.1088/1742-6596/2265/2/022086, 2022.

---

## Author Comment (AC2)

**Towards vertical wind and turbulent flux estimation with multicopter UAS**

Norman Wildmann[1] and Tamino Wetz[1]

[1]Deutsches Zentrum für Luft- und Raumfahrt e.V., Institut für Physik der Atmosphäre, Oberpfaffenhofen, Germany

**Correspondence:** Norman Wildmann (norman.wildmann@dlr.de)

**1 Review response**

We want to thank the reviewer for their careful and valuable review. We hope that we can clarify our analyses and clear out some of the concerns with our response.

**1.1 Review Comment 1**

1. *Given that most of the analysis is centered around optimal fits of field data, which is inherently noisy, it would be informative to include some analysis of the uncertainty of these fits and the sensitivity of the final wind estimation to these uncertainties. This would also help to alleviate typical concerns about optimal fits not being based on physical principals.*

   We understand that the noisy dataset for the vertical wind estimation is a critical point and needs to be analysed in more detail. In order to reduce the noise a bit further, we now apply a five-second moving average on the data before calibration. This improves the visual perception of the fits, but does not change the calibration coefficients.

   In order to evaluate the uncertainty we show in Fig. 1 the exponential fit as it has been applied in the first version of the manuscript, compared to a linear fit as suggested by reviewer #1. It shows that in the range of small vertical wind speeds ($w < 1\,\mathrm{m\,s^{-1}}$), very small differences appear, but for higher vertical wind speeds, especially downward winds, the linear fit seems to overestimate vertical winds significantly, although only few observations are there to prove it. The difference between the two fits (dashed and solid red line in Fig. 1b) over $F_z$ is shown in Fig. 2.

   For the derived average variances and fluxes in the presented dataset, the different fits have only a small influence, because most of the observed vertical winds are in the small wind speed range. Figure 3 shows the results of the validation (as in Fig. 8 of the manuscript) with a linear fit.

2. *The manuscript alludes to the fact that the wind-vane mode used in previous experiments was activated, but I don't see that explicitly stated anywhere.*

   Indeed, we are using a weather-vane mode. We did mention it in Wetz et al. 2021 and Wetz et al. 2022, but it is also true that we did not mention it in this manuscript. We agree that it is an important piece of information which we will include in the system description (Sect. 2.1) in the revised manuscript.

[Figure]

**Figure 1.** Fig. 5 (left) and Fig. 6 (right) with colorbars. The color coding of probability is limited to 2.0 and 0.5 respectively for best visualization of the scatter plot. All data points with values above are coded black. Different from the original manuscript, the dataset of vertical wind calibration (right) is smoothed with a 5 s windows and a linear fit is shown in addition to the exponential fit with a dashed red line.

[Figure]

**Figure 2.** Difference between exponential and linear fit for vertical wind speed.

3. *For the profiling UAS in the case study, are the same coefficients determined while hovering being used to estimate wind while ascending/descending? Bell et. al. (2020) showed that different coefficients are likely needed for an ascending UAS using a more rudimentary method for wind estimation. Would you expect the same here?*

   For our system, we did not observe significant differences in hovering and ascents with the UAS. We set the ascent rate comparatively low to $1\,\mathrm{m\,s^{-1}}$. We do not use descents, as the UAS has to fly through its own downwash and rotor-induced turbulence in that case.

4. *Was there any correction necessary to match up the UAS and sonic time series? If so, how was this done?*

   We are strictly using the system time of both systems without any further match up. Potential differences in the clocks are part of the uncertainty, but since both systems are synchronized to traceable clocks (GPS), we did not consider it necessary and could at least not observe any obvious bias in the data.

[Figure]

**Figure 3.** As Fig. 8 in the manuscript, but with linear fit of vertical wind speed.

5. *Were the winds from the sonic and UAS rotated into the mean wind independently of each other? In other words, was the sonic mean wind direction used to rotate to sonic winds, and vise versa? Or was one system used to determine the rotation for both?*

For calibration, we rotated both systems into the yaw angle of the UAS. For validation, the systems are individually rotated into the streamwise coordinate system. From Wetz et al. 2022 we know that the average uncertainty of wind direction as measured by the UAS is below 5°. With such small angle offsets, the uncertainties in the wind components depending on which information is used for the rotation is very small. We will include the information in a revised manuscript.

6. *Though small, there does seem to be a consistent bias in wind direction in Fig 11. Would the bias be attributable to GPS errors? Was there a magnetometer calibration for the flight site?*

As can be seen from Wetz et al. 2022, the uncertainty of UAS wind direction measurement compared to the mast is below 5°, but for individual systems it can of course occur that biases exist. The magnetometers of all UAS in the fleet were calibrated prior to the first flights of the campaign at the measurement site. As it can be seen in Fig. 11, the bias becomes largest with lowest wind speed, it is thus conceivable that the method to derive wind direction from yaw and the tangent of the streamwise and lateral wind component is prone to some error in such conditions. Throughout this morning transition, wind speeds are comparatively low.

7. *I recommend replacing 'unmanned aerial systems' with the more inclusive 'uncrewed aerial systems' or 'remotely piloted aircraft systems'.*

In some of my previous studies I used the term 'remotely piloted aircraft systems' (RPAS), because ICAO chose the term as the standard in their nomenclature. Most recently, in 2019, the EASA released their new regulations using the term 'Unmanned Aircraft Systems' to separate it from 'manned' aircraft, which is still the official term in aviation. For that reason, I switched to UAS. I appreciate the effort for inclusive language and the ideas behind it, but on the other hand, I also want to try to reduce the confusion in terminology and comply to the terms which are used by the responsible authorities.

8. *The way the axis labels on the figures are formatted could cause some confusion at first. For example, on Fig 4 "revolutions / s-1" could be interpreted as "revolutions per s-1" instead of "revolutions with units of s-1".*

It is actually not a confusion, but on purpose that the axis labels can be read as a math expression. The value divided by the unit only leaves the number on the axis. I know this is a matter of some dispute, but I am referring to the SI unit brochure that recommends treating units as mathematical entities and thus use the forward slash in tables and figures to separate variable names from units (https://www.bipm.org/documents/20126/41483022/SI-Brochure-9-concise-EN.pdf/2fda4656-e236-0fcb-3867-36ca74eea4e3, https://www.nist.gov/pml/special-publication-811/nist-guide-si-chapter-7-rules-and-style-conventions-expressing-values). Unless the reviewer or journal insists on a different style, I would like to keep the current style.

9. *Colorbars should be included on Figs 5 and 6.*

We include the colorbars in the revised manuscript as shown here in Fig. 1.

10. *On Fig 7, it may be informative to also plot the differences of the time series.*

Figure 4 shows the differences of the time series for the flight that was shown in Fig. 7 of the manuscript. With the original sampling rate, the results are very noisy and the strong gradients in vertical velocity which do not occur at the exactly same time at the location of the UAS and the sonic anemometer can cause large differences if the two time series are simply subtracted. Shown here is the time series with a 10-second moving average, but even here it shows that in periods with gradients as at 11:10 UTC, errors can become large. This can however not be interpret as an error of the

measurement method, but an uncertainty of the experimental setup. For average values, which we use for validation, this uncertainty is much smaller. For this reason, we suggest to not show such a plot in the manuscript. The time series comparison tells the story in a better way in our opinion.

[Figure]

**Figure 4.** Flight 115, UAS#13 with delta (right).

11. *On Fig 11, the errorbars are pretty difficult to see since they line up with the grid lines. It may be worth putting the cap on the end of the error bars to make them more visible.*

We put a cap on all the errorbars for better visibility in a revised manuscript as shown in Fig. 5.

[Figure]

**Figure 5.** Time series of horizontal wind speed $U$, wind direction $\Psi$, temperature $T$, friction velocity $u^*$ and TKE at 50 m above ground level during the morning transition of 28 June 2021 in comparison between UAS and sonic.

**References**

Wetz, T., Wildmann, N., and Beyrich, F.: Distributed wind measurements with multiple quadrotor unmanned aerial vehicles in the atmospheric boundary layer, Atmospheric Measurement Techniques, 14, 3795–3814, https://doi.org/10.5194/amt-14-3795-2021, 2021.

85  Wetz, T. and Wildmann, N.: Spatially distributed and simultaneous wind measurements with a fleet of small quadrotor UAS, Journal of Physics: Conference Series, 2265, 022 086, https://doi.org/10.1088/1742-6596/2265/2/022086, 2022.